# Pseudogenes in Cancer: State of the Art

**DOI:** 10.3390/cancers15164024

**Published:** 2023-08-08

**Authors:** Arturo Kenzuke Nakamura-García, Jesús Espinal-Enríquez

**Affiliations:** National Institute of Genomic Medicine, Mexico City 14610, Mexico; anakamura@inmegen.edu.mx

**Keywords:** pseudogene, gene expression regulation, tumorigenesis

## Abstract

**Simple Summary:**

Out of the billions of nucleotides comprising the human DNA, a substantial proportion (98%) represents non-coding DNA, meaning DNA that is not translated into proteins. Among the various types of non-coding DNA, pseudogenes stand out as duplicates of protein-coding genes that have undergone multiple alterations, rendering them unable to produce the protein they originally encoded. Despite their inability to generate functional proteins, recent studies have revealed the involvement of pseudogenes in several diseases, including cancer. In this review, we aim to provide a comprehensive overview of pseudogene formation, the mechanisms governing their expression, and the potential roles they may play in promoting tumorigenesis.

**Abstract:**

Pseudogenes are duplicates of protein-coding genes that have accumulated multiple detrimental alterations, rendering them unable to produce the protein they encode. Initially disregarded as “junk DNA” due to their perceived lack of functionality, research on their biological roles has been hindered by this assumption. Nevertheless, recent focus has shifted towards these molecules due to their abnormal expression in cancer phenotypes. In this review, our objective is to provide a thorough overview of the current understanding of pseudogene formation, the mechanisms governing their expression, and the roles they may play in promoting tumorigenesis.

## 1. Introduction

More than 98% of the human genome consists of non-coding DNA (ncDNA) [1]. These non-coding sequences can transcribe distinct types of RNA molecules, such as long non-coding RNAs (lncRNAs), microRNAs (miRNAs), transfer RNAs (tRNAs), among others. These molecules play different roles in the cell’s regulatory program, and any alterations to their function can impact the phenotype of an organism. Indeed, ncDNA sequences and their RNA products have been implicated in various biological processes, including human disorders [2,3,4,5,6,7,8]. Most of these non-coding DNA sequences can be readily distinguished based on their unique nucleotide sequences. However, there is a class of ncDNA with sequences highly similar to those of coding genes: the pseudogenes.

The term “pseudogene” was coined in 1977 by Jacq et al. [9] when they discovered a DNA sequence in *Xenopus laevis* that closely resembled the gene for 5S rRNA but exhibited various deteriorative characteristics. Subsequently, DNA sequences exhibiting a close resemblance to protein-coding genes, yet lacking apparent functional products, came to be referred to as pseudogenes derived from the respective coding genes [10]. Pseudogenes can originate from gene duplication events or retrotransposition processes, and often carry deleterious mutations that impede their transcription or translation, preventing the synthesis of functional peptides [10,11]. Initially categorized as “junk DNA” due to their perceived lack of function, recent studies have provided compelling evidence of their active involvement in normal tissue functioning as well as disease processes.

The prevailing perception of pseudogenes as non-functional entities, with the assumption that the mutations they accumulate are not subject to selective pressure, has been challenged by compelling evidence suggesting the contrary. Several studies have provided insights into the functional significance of pseudogenes by highlighting intriguing patterns in mutation frequencies and conservation across species.

For instance, in chickens, the occurrence of stop codon mutations in IglV and IghV pseudogenes, as well as in VH pseudogenes in mice, has been found to be lower than expected if mutations were random, indicating they could be under selective pressure [12,13]. Similarly, in the *Drosophila* Est-6 pseudogene, synonymous mutations were observed to occur more frequently than non-synonymous mutations, implying potential functional significance [14].

The conservation of certain pseudogenes across species adds to the growing evidence of their biological relevance. For example, Sudbrak et al. [15], analyzed the extended MHC class II region of the rhesus macaque and found two pseudogenes homologous to the human HIV TAT-specific factor-1-like and zinc finger-like pseudogenes. In a genome-wide survey comparing pseudogenes and their parental genes between humans and mice, 30 ancient pseudogenes shared between the two species were identified, suggesting their origin predates their speciation [16]. In another study, Khachane and Harrison [17], demonstrated that nearly 50% of transcribed pseudogenes in humans are conserved in the rhesus monkey, while only 3% are conserved in mice.

Collectively, the evidence of pseudogene conservation across diverse species suggests that they are subject to selective pressure and may serve biological functions despite carrying detrimental mutations. These findings challenge the notion of pseudogenes as mere genetic relics and highlight the need for further exploration of their functional implications in various biological processes.

Furthermore, a compelling line of evidence supporting the biological functionality of pseudogenes lies in their transcriptional activity. There is evidence that, despite the presence of deleterious mutations, pseudogenes undergo transcription in various physiological contexts. However, unraveling the precise mechanisms governing pseudogene activity and expression has been a challenge, partly due to the limited research dedicated to this area and the technical difficulties involved in accurately distinguishing pseudogenes from their parental genes [18]. Nonetheless, significant progress has been made in associating pseudogene expression with diverse biological processes and conditions.

For instance, the pseudogene HBBP1 has been shown to play a role in driving erythropoiesis by interacting with HNRNPA1, which is a heterogeneous nuclear ribonucleoprotein (hnRNP) that is known for promoting cytoplasmic RNA degradation [19,20]. TAL1, a master regulator of erythropoiesis, is a target of HNRNPA1, and thus expression of HBBP1 helps to stabilize TAL1, promoting erythropoiesis [21]. A systematic analysis conducted by Kalyana-Sundaram et al. [11] examined pseudogene transcription across multiple cancer and normal tissues. The study identified 154 tissue/lineage-specific pseudogenes, 848 moderately specific pseudogenes, and 165 ubiquitously expressed pseudogenes, primarily originating from housekeeping genes. Most pseudogenes were expressed in both normal and cancer tissues. Interestingly, 218 pseudogenes were found exclusively expressed in cancer tissues, with 178 observed in multiple types of cancers and 40 displaying high specificity for a single cancer type [11]. The growing interest in unraveling the potential role of pseudogenes in driving tumorigenesis has significantly contributed to the exploration of their functions, although a comprehensive understanding of their roles remains elusive.

Therefore, this review aims to provide a comprehensive summary and discussion of recent advances in understanding the regulation (or dysregulation) of pseudogene expression in cancer phenotypes. By synthesizing current research findings, we aim to describe current knowledge regarding the mechanisms underlying pseudogene involvement in tumorigenesis.

## 2. Classification of Pseudogenes

Pseudogenes have been categorized into three primary classes, providing a framework for understanding their origin, characteristics, and the mechanisms underlying their formation (Figure 1) [22,23]. These classes include processed, unprocessed (or duplicated), and unitary pseudogenes. Such classification offers valuable insights into the diverse nature of pseudogenes and the distinct processes by which they emerge.

### 2.1. Processed Pseudogenes

Processed pseudogenes originate from the retrotransposition of mature mRNA molecules back into the genome. This retrotransposition process involves the conversion of mRNA into DNA through reverse transcription, followed by integration into a new genomic location, often on a different chromosome [24,25] (see Figure 1A). For example, the pseudogene PPP1R26P1 is believed to have retroposed into the intron 2 of RB1 before the split between The New and Old World monkeys. This pseudogene evolved into a new promoter and initial exon for an alternative transcript of RB1 before the split of human and rhesus macaque [26].

Due to their origin from mRNA molecules, processed pseudogenes typically lack the regulatory sequences present in their parental genes [27]. Consequently, they are influenced by the regulatory elements surrounding their new genomic location. Processed pseudogenes lack introns and possess a poly-A tail, resembling the original mRNA from which they were derived. They are commonly flanked by direct repeats, a result of the target-primed reverse transcription (TPRT) process.

The formation of processed pseudogenes occurs through the activity of retrotransposons, such as LINE (Long Interspersed Nuclear Element) [28] or SINE (Short Interspersed Nuclear Element) [29] elements. These retrotransposons provide the necessary enzymatic machinery for reverse transcription and subsequent integration into the genome.

### 2.2. Unprocessed Pseudogenes

Unprocessed pseudogenes originate from the duplication of complete genomic regions, encompassing both exons and introns, of a functional gene. This duplication event can occur through mechanisms like unequal crossing over or tandem duplications (see Figure 1B).

For example, the duplicated pseudogene HBBP1 is believed to have originated in a common ancestor of placental mammals. In humans, HBBP1 seems to have evolved under functional constrains, as it shows signs of a slowdown in its exonic evolution, possibly because of its role (described above) in human erythropoiesis. This was not the case in other primates, were it seems to have evolved neutrally [21,30].

Unlike processed pseudogenes, unprocessed pseudogenes retain the intronic and intergenic regions of the parental gene, including essential regulatory elements like promoters and enhancers [31]. Their sequences exhibit high similarity to the original gene and are often located in close proximity to it. Unprocessed pseudogenes are formed through DNA-based duplication events that replicate an entire gene locus, along with its non-coding regions. Over time, the duplicated gene sequence may accumulate mutations that render it non-functional as a protein-coding gene.

It is worth noting that mammalian genomes primarily contain a significant proportion of processed pseudogenes compared to unprocessed pseudogenes. In contrast, plants exhibit an inverted pattern, potentially influenced by differences in the diversity and abundance of L1 subfamilies, which are the primary drivers of retrotransposition events [32].

### 2.3. Unitary Pseudogenes

Unitary pseudogenes emerge from functional genes through mutational events that result in their loss of functionality. These mutations can encompass frame-shift mutations, premature stop codons, or extensive deletions, ultimately abolishing their protein-coding capacity (see Figure 1C).

Such pseudogenization leads to the elimination of the gene from the genome, as there are no other functional copies present. However, it is possible to find coding orthologs of the gene in the genomes of other species [33]. Unitary pseudogenes often exhibit sequence variations compared to their functional gene counterparts due to the accumulation of aforementioned mutations. Consequently, they may display reduced or completely absent transcriptional activity.

Unitary pseudogenes typically exist as single copies within the genome, in contrast to the duplicated nature observed in unprocessed pseudogenes. For instance, the GULP locus in the human genome serves as an example of a unitary pseudogene that evolved from the L-gulono-gamma-lactone oxidase (GULO) gene. This enzyme plays a crucial role in ascorbic acid (vitamin C) biosynthesis and is present in the genomes of various vertebrates. However, the GULP locus has undergone multiple mutations that rendered the enzyme non-functional in humans. As our ancestors obtained sufficient ascorbic acid from their diet, the loss of this gene did not confer a disadvantageous trait [33,34].

### 2.4. Regulation of Pseudogene Expression

The regulation of pseudogenes involves intricate processes governing their transcription and translation, which are influenced by various factors. In the case of processed pseudogenes, they can integrate into the intronic regions of coding genes, effectively “hitchhiking” on the regulatory machinery of the host gene (see Figure 2A). Consequently, the retroposed gene can be transcribed as a fusion transcript with its host gene or as a splice variant, avoiding interference with the host gene’s normal functions [25,35].

Furthermore, processed pseudogenes can be modulated through their integration into open chromatin regions characterized by widespread transcription (see Figure 2B). This integration allows the retroposed element to be subjected to the regulatory elements of nearby genes, influencing its own expression pattern [25,35].

The regulation of retrocopies is also influenced by the transcription start site (TSS) of the parental gene (Figure 2C). If the parental gene has multiple TSSs, retrocopies can inherit promoter sequences from distant upstream TSSs, thereby acquiring the regulatory elements necessary for their own transcriptional control [36,37].

Additionally, retrocopies can be regulated through the influence of distant bidirectional promoters (Figure 2D) or CpG-rich proto-promoters that gain regulatory activity due to mutational events in their sequences (Figure 2E) [36].

In contrast, unprocessed or duplicated pseudogenes, arising from events such as tandem duplication or unequal crossing-over during meiosis, have the potential to retain the upstream regulatory elements of their parental genes (Figure 2F) [30,38].

In addition to their transcriptional activity, certain pseudogenes have the remarkable ability to undergo translation, giving rise to short peptides with functional significance [39]. Interestingly, even pseudogenes with truncated open reading frames (ORFs) are capable of producing biologically active peptides. A notable example is NOTCH2NL, which possesses less than half of the coding sequence of its parental gene, NOTCH2, and exclusively encodes the ligand-binding domain. Strikingly, NOTCH2NL can effectively inhibit Delta/Notch interactions by binding to Delta, thereby activating the Notch pathway [40]. This highlights the intriguing capacity of pseudogenes to exert physiological effects through peptide production, even with a fraction of the original coding capacity.

## 3. Functional Role of Pseudogenes

Pseudogenes play diverse functional roles, not only through their potential to generate peptides but also through their RNA molecules and DNA sequences, each employing distinct mechanisms to regulate the expression of other genes.

### 3.1. Pseudogenes as Endogenous Competitors

Since the high similarity to their parental genes, pseudogenes can function as endogenous competitors, also known as competitive endogenous RNAs (ceRNAs), by sequestering regulatory elements such as miRNAs or RNA-binding proteins (RBPs) (Figure 3) [41].

In terms of pseudogenes’ influence on transcriptional regulation of coding genes, Salmena et al. [42] proposed the concept of a cross-talk among mRNA, pseudogenes, and long non-coding RNAs (lncRNAs) through competition for common miRNA response elements (MREs) between the transcripts. These competitors are referred to as ceRNAs, and they affect gene expression by sequestering miRNAs capable of binding to both the ceRNA and mRNA targets (Figure 3). This hypothesis has provided a framework for exploring potential functions of lncRNAs and pseudogenes in different physiological contexts.

Competition for molecules other than miRNAs has also been reported, including RNA-binding proteins (RBPs) [41]. Initially, ceRNA activity was proposed as a mechanism by which pseudogenes regulate the expression of their parental genes. However, subsequent studies have identified potential ceRNA activity of pseudogenes that operates independently of their parental genes.

Competition for regulatory elements can impact physiological states by either promoting or repressing the expression of the competing RNAs. For instance, in Figure 3, we represent a model in which competition between a coding gene and a pseudogene is critical for maintaining the normal condition. In this example, a low concentration of the pseudogene promotes binding between the coding gene and the regulatory elements, which helps to maintain the normal physiological state. However, when the overexpression of a pseudogene occurs, it can hinder the binding between the coding gene and the regulatory elements, leading to an alteration in the physiological state.

### 3.2. Pseudogene-Mediated Hybridization with Coding Genes

Pseudogenes can form RNA-RNA duplexes with high complementarity to the sequences of other transcripts, regardless of their parental gene origin. These duplexes have the ability to block the translation of the target sequence or undergo processing to generate endogenous small interfering RNAs (esiRNAs). The formation of esiRNAs occurs through the cleavage of double-stranded RNA (dsRNA) by the Dicer protein. Alternatively, hairpin loops formed within single transcript homologous regions can also generate esiRNAs.

Upon cleavage by Dicer, the dsRNA is separated, and one strand is selected as the guide strand based on its thermodynamic stability. The guide strand is then incorporated into the RNA-induced silencing complex (RISC) and associated with a protein from the Argonaute family. Subsequently, RISC recognizes the target sequence through the formation of a perfect duplex, leading to the cleavage of the duplex at the middle position by the AGO protein through an endonucleolytic reaction [43] (Figure 4).

Through this mechanism, pseudogenes have been showed to influence gene expression in mouse oocytes [44,45] and *Trypanosoma brucei* [46]. Interestingly, in Tam et al. [44], multiple siRNAs were identified to resemble the sequence from HDAC1, however most of them were derived from pseudogenes of HDAC1.

### 3.3. Regulation of DNA Structure

Pseudogenes have the ability to impact chromatin structure through their transcripts or DNA sequences alone. Certain pseudogene transcripts have been identified as modifiers of epigenetic profiles by recruiting molecules involved in methylation and demethylation processes (Figure 5A). For example, XIST, a non-coding RNA crucial for the random silencing of one X chromosome in human females, is considered an intermediate pseudogene. It is believed to have originated from the pseudogenization of the protein-coding gene Lnx3 and various transposons. XIST recruits numerous chromatin remodelers and repressor complexes to mediate X chromosome inactivation [47,48,49,50,51].

Moreover, the DNA sequence of a pseudogene can induce DNA looping by interacting with the sequence of another gene (Figure 5B). For instance, the expression of human globin genes is regulated by an upstream locus control region (LCR) that governs gene expression in a developmentally specific manner. It has been proposed that the pseudogene HBBP1, located in the intergenic region between the Aγ- and δ-globin genes, participates in long-range interactions to shape DNA structure. These interactions facilitate contact between the LCR region and its target genes during specific developmental stages, promoting the transition of globin expression from the fetal to adult stages [52].

Pseudogene sequences can also influence DNA through gene conversions, wherein a portion of a pseudogene sequence can be inserted into the sequence of its parental gene, potentially introducing mutations (Figure 5C). This mechanism allows pseudogenes to serve as a reservoir for sequence diversity and can contribute to the diversification of proteins, such as immunoglobulins [22,53].

## 4. Pseudogenes as Functional Molecules in Cancer

Previous works have associated the expression of pseudogenes with different types of cancers (for a through revision, please see the works of Sisu [54] and Stasiak et al. [55]). In this section, we will discuss several examples of these cancer associated pseudogenes and explore their potential action mechanism. Table 1 and Table 2 provide a summary of the effects of different pseudogene expressions in cancer, including both promoting (Table 1) and suppressing (Table 2) tumorigenesis.

### 4.1. Pseudogenes as ceRNAs in Cancer

In hepatocellular carcinoma (HCC), the aberrant activation of SNRPFP1 has been observed, correlating with worse clinical-pathologic features and promoting processes such as cell proliferation, apoptosis resistance, and cell motility [56]. Interestingly, SNRPFP1 expression is inversely correlated with miR-125-5p, a known tumor-suppressive transcript in various cancers [57]. Experimental evidence suggests that SNRPFP1 may act as a competitive endogenous RNA (ceRNA) by sequestering miR-125-5p through binding sites on its 3′ end, thus reducing the suppressive effect of this miRNA and promoting HCC progression [56].

In a study by Carron et al., an increased risk of developing oropharynx squamous cell carcinoma was associated with the presence of three or more copies of the pseudogenes ADAM3A and ADAM5 [58]. The authors also identified a highly homologous region in the 3′-UTR sequence of ADAM5 and ADAM9 that serves as a binding site for miR-122b-5p. The competition for this miRNA binding site between ADAM5 and ADAM9 suggests a potential mechanism for the coordination of their expression and the promotion of tumorigenesis [58].

PTTG3P, detected in various cancer types such as colorectal, tongue, and prostate cancer, shares high homology with the 3′ UTR of JAG1, an oncogenic protein that activates the NOTCH pathway. In oral cancer, PTTG3P acts as a ceRNA for JAG1 by sponging miR-142-5p, leading to increased JAG1 translation and enhanced cancer cell proliferation [59]. In castration-resistant prostate cancer, PTTG3P upregulation confers resistance to androgen-deprivation therapy through competition with PTTG1 for miR-146a-3p binding, thereby modulating PTTG1 expression [60].

Vascular endothelial growth factor receptor-1 (VEGFR1 or FLT1) plays a role in promoting epithelial-mesenchymal transition and an aggressive phenotype in cancer cells. Ye et al. investigated the regulation of VEGFR1 by its pseudogene FLT1P1 in colorectal cancer cells [61]. Their findings revealed that FLT1P1 exhibits bidirectional transcription, producing both sense and antisense transcripts. The sense molecule (FLT1P1-s) enhances VEGFR1 protein expression, while the antisense transcript (FLT1P1-as) has the opposite effect by impairing its translation. FLT1P1-as downregulates VEGFR1 and its ligand, VEGF-A, by interacting with miR-520a, leading to inhibition of cell proliferation and tumor growth [61].

### 4.2. Pseudogenes as Cancer Markers

Pseudogenes can also serve as valuable markers for cancer. For instance, Zhu et al. identified TCAM1P as highly and specifically expressed in cervical cancer, with its expression being dependent on human papillomavirus (HPV) infection [62]. The RNA-binding protein EIF4A3 was found to stabilize TCAM1P expression, highlighting the intricate regulatory network involving pseudogenes in cancer.

MYLKP1, a partially duplicated gene derived from MYLK. MYLK exhibits diverse alternative splicing patterns that generate nine different transcripts [63]. Interestingly, MYLKP1 retains a promoter sequence highly homologous to the promoter of smooth muscle myosin light chain kinase (smMLCK), a product of MYLK [64]. Han et al. demonstrated that MYLKP1 expression promotes cell proliferation in cancer cells, with MYLKP1’s promoter activity being increased in lung adenocarcinoma cells, comparable to the smMLCK promoter [64]. Moreover, MYLKP1 was found to suppress the mRNA and protein expression of smMLCK, potentially through competition for an RNA-binding protein and destabilization of the MYLK transcript [65].

DUXAP10, overexpressed in multiple cancer types including pancreatic, gastric, and colorectal cancer, has been associated with cell proliferation, disease progression, and lymph node metastasis [66,67,68]. Its increased expression highlights its potential as a cancer marker and its involvement in tumorigenesis.

Beyond aberrant pseudogene expression, an additional mode of regulation involves the immune response triggered by short peptides derived from pseudogenes. The translation of small amino acid sequences from pseudogenes expressed in cancer cells can elicit an immune response against malignant cells. Notably, peptides derived from pseudogenes such as HSD17B12 and NA88-A have been detected on the surface of cancer cells [69,70]. These pseudogene-derived peptides can contribute to the immune surveillance of cancer cells, providing a potential avenue for immunotherapeutic strategies [71,72].

### 4.3. Pseudogene Hybridization in Cancer

In hepatocellular carcinoma (HCC), a comprehensive analysis of actively transcribed pseudogenes revealed the presence of 448 pseudogenes capable of producing endogenous small interfering RNAs (esiRNAs) that regulate protein-coding genes (Figure 4) [73]. Among these pseudogenes, a particular one originated from protein phosphatase 1K, mitochondrial (ψPPM1K), was identified as having the potential to regulate multiple protein-coding genes. The study demonstrated the regulatory impact of siRNAs derived from ψPPM1K, which effectively targeted its cognate gene and resulted in the inhibition of cell growth by downregulating NEK8 expression. Notably, the regulatory effects were primarily attributed to a specific siRNA generated from a hairpin structure formed by inverted repeats within the pseudogene’s RNA sequence [73].

### 4.4. Pseudogenes Altering DNA Structure in Cancer

Pseudogenes have been observed to modify chromatin structure through their transcripts or DNA sequences, and their involvement in cancer has been documented. In hepatocellular carcinoma (HCC), a gene conversion event between CYP2A6 and its pseudogene (CYP2A7) results in the CYP2A61B polymorphism, which carries a fragment from CYP2A7 in the 3′ UTR region. This polymorphism exhibits increased stability and enhanced enzyme activity. Interestingly, individuals homozygous for CYP2A61B were found to have a higher cigarette consumption, potentially increasing the risk of lung cancer due to elevated nicotine metabolism [74,75].

DUXAP8 has been implicated in various cancer types and contributes to tumorigenesis through different mechanisms. In non-small-cell lung cancer, DUXAP8 epigenetically regulates EGR1 and RHOB. Its expression has been associated with worse clinical features, and knockdown of DUXAP8 reduces cancer cell growth and survival, leading to increased expression of EGR1 and RHOB transcripts. Notably, DUXAP8 has been shown to interact with several RNA-binding proteins (RBPs), particularly LSD1 and EZH2, which are negative transcriptional regulators involved in epigenetic modifications. Chromatin immunoprecipitation (ChIP) analysis revealed that LSD1 and EZH2 bind to the promoters of RHOB and EGR1, respectively, and DUXAP8 disrupts their binding and epigenetic modification capabilities. Furthermore, DUXAP8 overexpression has been detected in pancreatic cancer cells, and its suppression leads to reduced cell growth and inhibition of tumor growth, accompanied by increased expression of multiple genes involved in tumor suppression [76,77,78,79,80,81].

SALL4, a stem cell factor associated with embryogenesis and stem-like tissues, exhibits aberrant expression in various malignancies. In hepatocellular carcinoma, the binding of SALL4 pseudogenes to DNMT1 and their impact on SALL4 expression have been investigated. A negative correlation was found between SALL4 expression and methylation levels in a specific region of the 5′ UTR-exon 1. Methylation analysis of different HCC cell lines confirmed that the methylation profile plays a critical role in SALL4 expression. Notably, SALL4 expression was higher in cells with a hypomethylated profile. Blocking DNMT1 in cells with a methylated profile resulted in SALL4 upregulation and enhanced cellular growth. Additionally, transient overexpression of distinct SALL4 pseudogenes in cells with a methylated profile led to demethylation of the 5′ UTR-exon 1-intron 1 CpG island and upregulated SALL4 expression. Knockdown of the SALL4 pseudogene in cells with a demethylated profile increased the methylation status, and interaction analysis demonstrated the binding of the SALL4 pseudogene transcript with DNMT1. These findings suggest that the pseudogene, SALL4P5, may be responsible for the hypomethylation of the SALL4 promoter CpG region by interacting with DNMT1 [82].

**Table 1 cancers-15-04024-t001:** Pseudogenes with reported tumor-promoter effects.

Pseudogene	Related Cancer	Action Mechanism	Ref
ADAM5	Oropharynx squamous cell carcinoma	Upregulates ADAM9 by binding miR-122b-5p	[58]
ACTG1P25	Breast cancer	Its upregulation promotes endocrine therapy resistance by competing with E2F1 for binding to PURA, thereby increasing E2F1 expression and activating cell cycle-related genes	[83]
AK4P1	Pancreatic cancer	Upregulates genes related with cell proliferation through binding with tumor suppressive miR-375	[84,85,86]
BRAFP1	Thyroid tumors, among others	Elevate BRAF expression and MAPK activation through ceRNA mechanism	[87,88]
BRCA1P1	Breast and ovarian cancer	A recombination between BRCA1P1 and BRCA1 can remove the promoter and initiation codon of BRCA1, thus blocking its tumor-suppressive functions	[89]
CYP2A7	Lung cancer	Gene conversion with its parental gene (CYP2A6) produces a polymorphism with enhanced nicotine metabolism and associated with an increased cigarette consumption	[74,75]
CYP4Z2P	Breast cancer	The ceRNA network of CYP4Z1 and pseudogene CYP4Z2P inhibit apoptosis in cancer cells by sharing miRNA miR-125a-3p binding sites	[90]
DUXAP8	Lung and pancreatic cance	Contributes to cancer progression by recruitment of epigenetic machinery to silence tumor suppressive genes	[80,81]
EBLN3P	Non-small cell lung cancer, osteosarcoma and colorectal cancer	Promotes cancer cell proliferation and epithelial-mesenchymal transition by sponging of different miRNAs	[91,92,93,94]
FTH1P3	Non-small cell lung cancer	Recruits LSD1 to epigenetically downregulate TIMP3, promoting tumor malignancy	[95]
FLT1P1-S	Colorectal cancer	Positive regulator of VEGFR1 expression (opposite regulatory function from FLT1P1-AS)	[61]
OGFRP1	Gastric cancer	Suppresses cell apoptosis by regulating the miR-149-5p/MAP3K3 axis.	[96]
LGMNP1	Glioblastoma	Promotes aggresiveness of cancer cells by downregulating miR-495-3p, possibly through a RISC related mechanism	[97]
MSTO2P	Colorectal cancer	Promotes cancer cell proliferation by epigenetically downregulating CDKN1A through binding with EZH2	[98]
MYLKP1	Lung cancer	MYLKP1 is overexpressed in cancer cells and downregulates smMLCK, possibly by decresing its stability through competition for RBPs	[64,65]
OCT4-pg4	HCC	Upregulates OCT4 by sequestering miR-145, promoting tumorigenicity	[99,100]
PCNAP1	Breast cancer	Promotes invasion of cancer cells by binding with miR-340-5p, hence upregulating SOX4	[101]
PDIA3P1	Glioma	Sequestrates miR-124-3p to upregulate RELA expression, promoting glioma cells MES transition by activating the NF-κB pathway	[102]
PPM1K	HCC	Produces esiRNAs that downregulates NEK8, inhibiting cell growth	[73]
PRELID1P6	Glioma	Promotes cancer cell proliferation by upregulating hnRNPH1 and TRF2, which activates the Akt/mTOR pathway. It is downregulated by miR-1825.	[103]
PTENP1-AS	Melanoma	Recruits EZH2 and H3K27me3 to downregulate PTEN expression	[104]
PTTG3P	Oral and prostate cancer	Functions as ceRNA by binding with miR-142-5p (oral) and miR-146a-3p (prostate)	[59,60]
RPSAP52	Breast cancer and sarcoma cell lines	Contributes to cancer progression by controlling the HMGA2/IGF2BP2/LIN28B axis and downregulating et-7 miRNAs	[105]
SALL4P5	HCC	Demethylates SALL4 by interacting with DNMT1	[82]
TCAM1P	Cervical cancer	Regulates cell cycle and promotes cancer cell proliferation, its expression is HVP-dependent and is regulated by HPV E6/E7 and EIF4A3	[62]
TDGF1P3	Colorectal cancer	Upregulates PKM2 by competing and binding with miR-338-3p	[106]
RP9P	Colorectal cancer	Promotes colorectal cancer progression through upregulation of FOXQ1 by competing for miR-133a-3p	[107]
UBE2CP3	Gastric cancer	Promotes gastric cancer progression by upregulating ITGA2 through binding with miR-138-5p	[108]

**Table 2 cancers-15-04024-t002:** Pseudogenes with reported tumor-suppressive effects.

Pseudogene	Related Cancer	Action Mechanism	Ref
ARHGAP27P1	Gastric cancer	Exerts tumor-suppressive functions by interacting with JMJD3 and epigenetically activation of p15, p16 and p57	[109]
FLT1P1-AS	Colorectal cancer	Downregulates VEGFR1 and VEGF-A by interacting with miR-520a and by blocking VEGFR1 translation	[61]
FOXO3P	Breast cancer	Suppresses tumor growth by binding with multiple miRNA, thus upregulating FOXO3 mRNA	[110]
Pseudogenes of FTH1	Prostate cancer	The ceRNA networks formed by FTH1 and their pseudogenes exerts a tumor suppressive effect by bind with multiple miRNAs	[111]
GUSBP11	Triple negative breast cancer	Inhibits cancer progression by upregulating SPNS2 through sequestering miR-579-3p	[112]
MT1JP	Triple negative breast cancer	Inhibits TNBC by regulating the miR-138/HIF-1α axis	[113]
PEBP1P2	Clear cell renal cell carcinoma	Exerts tumor suppressive effects by recruiting the YBX1/ELAVL1 complex to stabilize PEBP1; additionally, can act as a ceRNA for KLF13 sponging several miRNAs	[114]
SNRPFP1	HCC	Binds with miR-125-5p blocking its tumor suppressive action	[56]
TUSC2P	Esophageal squamous cell carcinoma	TUSC2P 3′UTR serves as a decoy to protect TUSC2 from binding with miR-17-5p, miR-520a-3p, miR-608, miR-661, thus upregulating its translation and inhibiting cancer cell survival and proliferation	[115,116]

## 5. Co-Expression of Pseudogenes

Gene co-expression networks (GCNs) provide a global view of the transcriptome profile by analyzing the correlation between gene expression patterns [117,118]. Typically, this data is derived from next-generation sequencing experiments, such as RNA-seq. GCNs have emerged as valuable tools for investigating gene regulation in cancer [119]. Consequently, exploring the potential of GCNs to elucidate the role of pseudogenes in cancer is highly intriguing.

Pseudogenes, acting as competing endogenous RNAs, can modulate the availability of miRNAs, thereby influencing the regulatory effects on target genes. Co-expression networks offer the opportunity to identify interactions between pseudogenes and other molecules in specific contexts. These networks have previously shed light on the functional role of other non-coding RNAs, such as miRNAs [120,121].

Pseudogenes often exhibit similar expression patterns to their parental genes or related genes. Analyzing GCNs allows for the identification of co-regulated gene clusters or modules with coordinated expression patterns, including pseudogenes [122,123,124]. This approach has the potential to reveal regulatory relationships between pseudogenes and their associated genes, providing valuable insights into their functional roles.

For example, Carron, Coletta, and Lourenço conducted an analysis focused on GCNs of pseudogenes in a head and neck cancer (HNC) dataset [125]. By employing a pipeline designed to identify cancer-relevant pseudogene interactions, they discovered several modules comprising protein-coding genes and pseudogenes. Notably, these modules contained genes associated with carcinogenesis, cell cycle regulation, and immune response. In one instance, the cell cycle module included the pseudogene DNM1P47 and the gene TP53, suggesting an indirect interaction between these transcripts in HNC [125].

In another study, Chang et al. utilized ceRNA and gene co-expression networks to identify hub lncRNAs and pseudogenes associated with lung cancer [126]. Among the hundreds of detected interactions, one particular pair, the negative correlation between PVT1 and miR-423-5p, had been previously reported in thyroid cancer and was experimentally validated using dual-luciferase reporter, RNA immunoprecipitation, and RNA pull-down assays [127]. Further investigations that validate gene co-expression network interactions will undoubtedly provide new insights into the global regulation of gene expression.

Our group has previously focused on analyzing GCNs in cancer phenotypes to gain insights into the widespread changes in normal regulatory circuits that promote and sustain carcinogenesis. In multiple cancer gene expression profiles, we consistently observed a notable feature: a higher proportion of interactions between genes located on the same chromosome (intra-chromosome interactions) compared to the GCNs of their normal counterparts [128,129,130,131].

Although this phenomenon has been reported in various tissues, our recent study on the GCNs of hematopoietic cancer specifically examined the co-expression profile of pseudogenes in these cancer types [132]. Confirming the loss of inter-chromosomal regulation in hematopoietic cancers, we identified an increased proportion of interactions between pseudogenes, particularly those derived from housekeeping genes such as eukaryotic elongation factors and riboproteins. Furthermore, many of these pseudogenes exhibited higher expression levels compared to normal tissues. However, due to the limited understanding of these pseudogenes, we could derive only minimal biological insights from this analysis.

The co-expression analysis of gene expression profiles is a valuable approach for detecting potential rewiring of normal circuits contributing to the regulatory programs of diseases. Future research focused on understanding the functions of pseudogenes, particularly those derived from housekeeping genes, could provide valuable information regarding the biological significance of increased pseudogene co-expression observed in cancer GCNs [132]. Such investigations hold the potential to enhance our understanding of pseudogenes and their role in cancer.

## 6. Conclusions and Future Perspectives

The study of non-coding sequences, such as pseudogenes, presents a promising avenue for unraveling new genetic regulatory mechanisms, considering the substantial fraction of the human DNA that does not encode proteins. While many non-coding regions remain unexplored, it is evident that the notion of pseudogenes as “junk” or non-functional DNA is misguided. Recent evidence highlights their integral role in diverse biological processes, firmly establishing their inclusion in the broader genetic regulatory landscape. Further investigation into the non-coding regions of the genome is essential to uncover the functions, if any, associated with these regions.

The mechanisms employed by pseudogenes to influence gene expression are remarkably multifaceted, as indicated by the literature reviewed in this study. These mechanisms encompass the regulation of DNA structure and competition for regulatory elements with other genes. Notably, pseudogenes exhibit the ability to regulate not only the expression of their parental genes but also seemingly unrelated genes, revealing their extensive regulatory reach.

We have presented arguments demonstrating that pseudogene expression is not only significant for normal biological processes but also crucial for comprehending the complexity of genetic dysregulation in cancer. Like protein-coding genes, pseudogenes undergo evident deregulation in cancer, characterized by abnormal expression patterns and co-expression relationships. This observation implies the existence of an unexplored layer within the genetic regulatory program that involves pseudogenes. Moreover, this layer of regulation is profoundly disrupted during carcinogenesis, underscoring the potential of studying pseudogenes as a promising research area for advancing our understanding of the disease and developing novel therapeutic strategies.

## Figures and Tables

**Figure 1 cancers-15-04024-f001:**
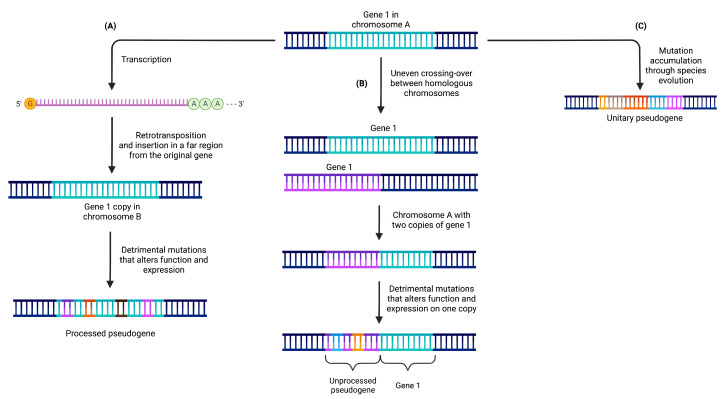
Pseudogenes can be categorized into three distinct classes based on the mechanisms responsible for their formation. (**A**) Processed pseudogenes originate from the retrotransposition of mRNA molecules and are typically located on a different chromosome compared to their parental gene. (**B**) Unprocessed pseudogenes are generated through gene duplication events followed by subsequent deleterious mutations that impede the expression of one of the duplicated copies. (**C**) Unitary pseudogenes arise from the accumulation of detrimental mutations in a gene that lacks any other copies in the genome, resulting in the complete loss of the functional gene.

**Figure 2 cancers-15-04024-f002:**
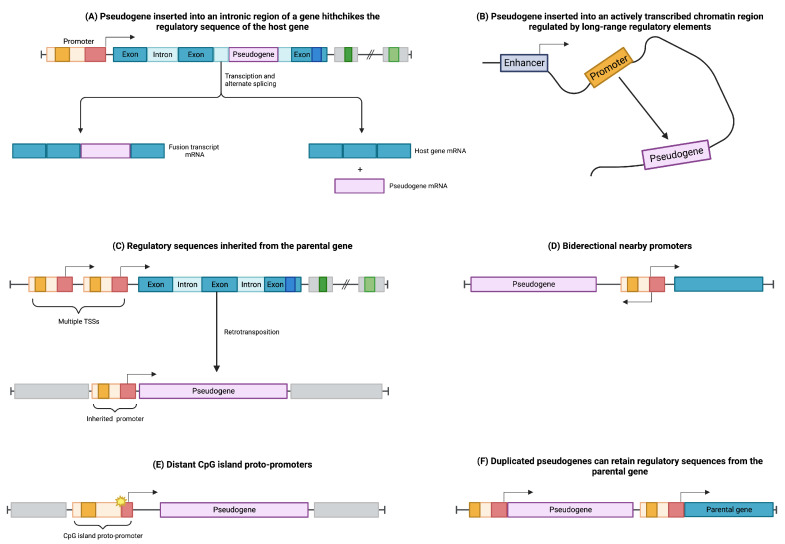
Modulation of pseudogene transcription involves distinct mechanisms. (**A**) Pseudogenes inserted into intronic regions of genes can exploit the regulatory machinery of their host genes. (**B**) Pseudogenes inserted into open chromatin regions can be influenced by long-range regulatory elements. (**C**) Pseudogenes derived from genes with multiple transcription start sites (TSSs) can inherit regulatory sequences. (**D**) Pseudogenes can be regulated by bidirectional promoters that control different genes. (**E**) Distant CpG islands can acquire promoter activity through sequence mutations, thereby regulating nearby pseudogenes. (**F**) Duplicated pseudogenes can retain regulatory elements from their parental genes.

**Figure 3 cancers-15-04024-f003:**
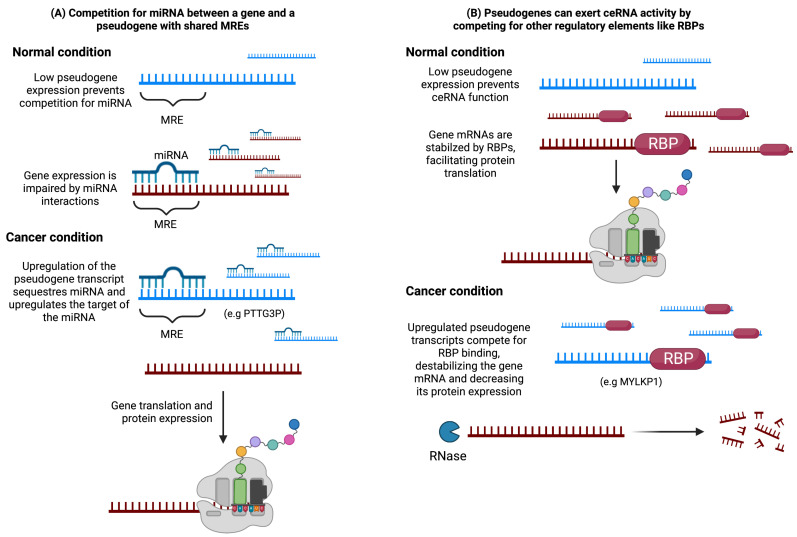
Pseudogenes play a crucial role in modulating gene expression through their involvement in ceRNA networks, which has been linked to cancer-related malignant features. (**A**) In ceRNA networks, pseudogene transcripts and mRNA molecules containing shared miRNA response elements (MREs) compete for the same pool of miRNAs. Under normal conditions with low pseudogene transcription, a gene with proto-oncogenic effects is effectively suppressed by the miRNA. However, during tumorigenesis, aberrant pseudogene expression can disrupt the regulation of the coding gene by sequestering the miRNA, leading to increased expression and enhanced oncogenic effects of the coding gene. (**B**) Another mechanism involves competition between the pseudogene and the coding gene for stabilization through binding with an RNA-binding protein (RBP). If the coding gene possesses tumor suppressive properties, the competition with the pseudogene can inhibit its translation and promote tumorigenesis.

**Figure 4 cancers-15-04024-f004:**
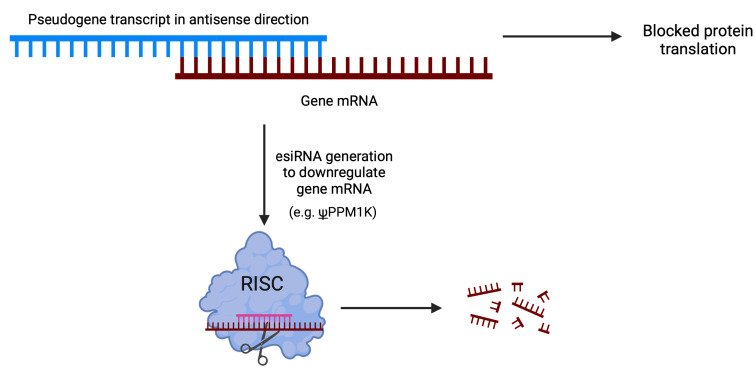
Pseudogenes can form RNA-RNA duplexes that can block mRNA translation and generate siRNAs to decrease mRNA target levels.

**Figure 5 cancers-15-04024-f005:**
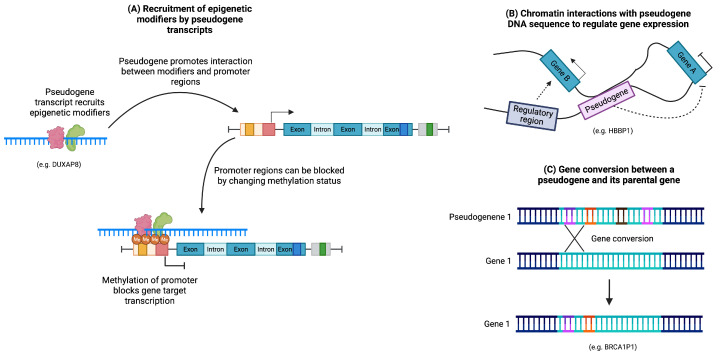
Regulation of DNA structure by pseudogenes. (**A**) Pseudogene transcripts can recruit epigenetic modifiers to modulate gene expression. (**B**) Pseudogene sequences can generate DNA loops that block or promote gene expression. (**C**) Gene conversions can introduce mutations into the sequence of its parental gene.

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
