# Peer review of "Pseudogenes in Cancer: State of the Art"

_cancers, 2023, doi:10.3390/cancers15164024_

Round 1

Reviewer 1 Report

In this manuscript, authors focus on the functional study of pseudogenes in cancer.  They summarized the genetic regulation mechanisms and function roles of pseudogenes in tumorigenesis.

Overall, this is a well written review worthy of publication. I don’t have any critiques.

Author Response

Reviewer 1

In this manuscript, authors focus on the functional study of pseudogenes in cancer.  They summarized the genetic regulation mechanisms and function roles of pseudogenes in tumorigenesis.

Overall, this is a well written review worthy of publication. I don’t have any critiques.

We want to thank the Reviewer for their positive comments. We have included some modifications based on the comments of the others revisions. 

Reviewer 2 Report

1. The therapeutic targets based on pseudogenes could be summarized. "doi: 10.1007/978-1-0716-1503-4_20."

2. The differences between pseudogenes and other ncRNAs could be discussed and summarized at the beginning.

Author Response

The therapeutic targets based on pseudogenes could be summarized. "doi: 10.1007/978-1-0716-1503-4_20."

ANS: The reference has been included in the revised version of this manuscript (line 247)

“(for a through revision, please see the works of Sisu [55] and Stasiak et al. [56])”

The differences between pseudogenes and other ncRNAs could be discussed and summarized at the beginning.

ANS: Thank you for this comment. The first paragraph has been modified to briefly comment on other types of ncRNAs as well as their relevance in biological processes:  

The therapeutic targets based on pseudogenes could be summarized. "doi: 10.1007/978-1-0716-1503-4_20."

ANS: The reference has been included in the revised version of this manuscript (line 247)

“(for a through revision, please see the works of Sisu [55] and Stasiak et al. [56])”

The differences between pseudogenes and other ncRNAs could be discussed and summarized at the beginning.

ANS: Thank you for this comment. The first paragraph has been modified to briefly comment on other types of ncRNAs as well as their relevance in biological processes:  

More than 98% of the human genome consists of non-coding DNA (ncDNA) [1]. These non-coding sequences can transcribe distinct types of RNA molecules, such as long non-coding RNAs (lncRNAs), microRNAs (miRNAs), transfer RNAs (tRNAs), among others. These molecules play different roles in the cell’s regulatory program, and any alterations to their function can impact the phenotype of an organism. Indeed, ncDNA sequences and their RNA products have been implicated in various biological processes, including human disorders [2 8]. Most of these non-coding DNA sequences can be readily distinguished based on their unique nucleotide sequences. However, there is a class of ncDNA with sequences highly similar to those of coding genes: the pseudogenes.” 

Reviewer 3 Report

Review of the manuscript entitled Pseudogenes in cancer: State of the Art.

Nowadays, the pseudogenes have become the focus of cancer research. The first review was published in the Nature Reviews Genetics (Cheetham et al., 2019), which gave a comprehensive view about the functions of pseudogenes. Two years later it was updated by Stasiak et al. (2021; Life, MDPI). Here it is the newest update of the achievement of the field.

Every review could add some novelty to the topic. The strength of the present review is the overview of the regulations of pseudogene expressions. The Figures are clear and easy to understand.

Minor flaws:

Please cite Stasiak et al., 2021; Life, MDPI.

Update the known pseudogenes implicated in cansers, that were already 37 two years ago according to Stasiak et al. Here, there were mentioned less than 30 genes.

There are only a few misspelling error.

Author Response

Review of the manuscript entitled Pseudogenes in cancer: State of the Art.

Nowadays, the pseudogenes have become the focus of cancer research. The first review was published in the Nature Reviews Genetics (Cheetham et al., 2019), which gave a comprehensive view about the functions of pseudogenes. Two years later it was updated by Stasiak et al. (2021; Life, MDPI). Here it is the newest update of the achievement of the field.

Every review could add some novelty to the topic. The strength of the present review is the overview of the regulations of pseudogene expressions. The Figures are clear and easy to understand.

We want to thank the Reviewer for their valuable and positive comments.

Minor flaws:

Please cite Stasiak et al., 2021; Life, MDPI.

ANS: The reference has been added in line 247: 

“(for a through revision, please see the works of Sisu [55] and Stasiak et al. [56])”

Update the known pseudogenes implicated in cansers, that were already 37 two years ago according to Stasiak et al. Here, there were mentioned less than 30 genes.

ANS: Tables 1 and 2 were updated with the relevant information from Stasiak et al., 2021. Note that only pseudogenes with known molecular action mechanisms are included in the tables. The aim of our manuscript is to explore the possible action mechanism of pseudogenes in order to facilitate hypothesis generation on the function of unexplored pseudogenes. However, many of the reported pseudogenes on Stasiak do not present a clear action mechanism through which they can influence tumorigenesis. Based on the report from Stasiak et al., we added four pseudogenes to our tables, namely PCNAP1, LGMNP1, PDIA3P1 (Table 1) and TUSC2P (Table 2). Additionally, we have included the pseudogene ACTG1P25 to Table 1, since a new report  of its involvement in promoting therapy resistance in breast cancer was just published this month. (DOI: 10.1158/0008-5472.CAN-23-0015). Thank you for pointing this out.

Reviewer 4 Report

Arturo Kenzuke Nakamura-García and Jesús Espinal-Enríquez presented a review focusing on pseudogenes and their role in cancer.

After a general discussion on the structure and classification of pseudogenes, the authors reported the molecular mechanisms through which they influence the expression of other genes. Finally, the authors presented examples of the involvement of pseudogenes in cancer. Painstaking work went into creating tables 1 and 2, listing the pseudogenes and their mechanism of action in cancer.

The review is clearly designed, and some recommendations are given below:

Line 54-56, unclear sentence, please make it clearer.

Paragraphs 2.1 and 2.2, list some examples, as done in paragraph 2.3.

Paragraph 3.1. This paragraph includes some concepts that are not pertinent to the subject matter, “pseudogenes as endogenous competitors”, in particular from line 165 to line 172. Furthermore, figure 3 describes two conditions, the “normal condition”, and “cancer condition”, not described in paragraph 3.1. It is recommended a rearrange the paragraph appropriately.

Paragraph 3.2 and Figure 4. The authors can present an example of a pseudogene with the function described in the paragraph. Similarly, Figure 4 shows the example of PPM1K, which however is not described in the paragraph. Please, rearrange the paragraph appropriately.

Paragraph 4.3, lines 285-288, sentence unclear, please make it clearer.

Author Response

Arturo Kenzuke Nakamura-García and Jesús Espinal-Enríquez presented a review focusing on pseudogenes and their role in cancer.

After a general discussion on the structure and classification of pseudogenes, the authors reported the molecular mechanisms through which they influence the expression of other genes. Finally, the authors presented examples of the involvement of pseudogenes in cancer. Painstaking work went into creating tables 1 and 2, listing the pseudogenes and their mechanism of action in cancer.

The review is clearly designed, and some recommendations are given below:

Thank you for these comments. In what follows, we will present a point-by-point response to all concerns and recommendations raised by the Reviewer.

Line 54-56, unclear sentence, please make it clearer.

ANS: The paragraph has been modified: “For instance, the pseudogene HBBP1 has been shown to play a role in driving erythropoiesis by interacting with HNRNPA1, which is a heterogeneous nuclear ribonucleoprotein (hnRNP) that is known for promoting cytoplasmic RNA degradation [19,20]. TAL1, a master regulator of erythropoiesis, is a target of HNRNPA1, and thus expression of HBBP1 helps to stabilize TAL1, promoting erythropoiesis [21].”

Paragraphs 2.1 and 2.2, list some examples, as done in paragraph 2.3.

ANS: As examples we added PPP1R26P1 to section 2.1 and HBBP1 to section 2.2:

Section 2.1: “For example, the pseudogene PPP1R26P1 is believed to have retroposed into the intron 2 of RB1 before the split between The New and Old World monkeys. This pseudogene evolved into a new promoter and initial exon for an alternative transcript of RB1 before the split of human and rhesus macaque [26].”

Section 2.2: For example, the duplicated pseudogene HBBP1 is believed to have originated in a common ancestor of placental mammals. In humans, HBBP1 seems to have evolved under functional constrains, as it shows signs of a slowdown on its exonic evolution, possibly because of its role (described above) in human erythropoiesis. This was not the case on other primates, were it seems to have evolved neutrally [21,30].”

Paragraph 3.1. This paragraph includes some concepts that are not pertinent to the subject matter, “pseudogenes as endogenous competitors”, in particular from line 165 to line 172. Furthermore, figure 3 describes two conditions, the “normal condition”, and “cancer condition”, not described in paragraph 3.1. It is recommended a rearrange the paragraph appropriately.

ANS: The reviewer is correct, lines 165 to 172 have been eliminated as they are not relevant for the section. A new paragraph is added to further explain Figure 3:

“Competition for regulatory elements can impact physiological states by either promoting or repressing the expression of the competing RNAs. For instance, in Figure 3, we represent a model in which competition between a coding gene and a pseudogene is critical for maintaining the normal condition. In this example, a low concentration of the pseudogene promotes binding between the coding gene and the regulatory elements, which helps to maintain the normal physiological state. However, when the overexpression of a pseudogene occurs, it can hinder the binding between the coding gene and the regulatory elements, leading to an alteration in the physiological state.”

Paragraph 3.2 and Figure 4. The authors can present an example of a pseudogene with the function described in the paragraph. Similarly, Figure 4 shows the example of PPM1K, which however is not described in the paragraph. Please, rearrange the paragraph appropriately.

  ANS: Please note that description of function of ψPPM1K pseudogene is provided in section 4.3. Figure 4 is modified to include the ψ symbol. We have added a paragraph to comment on reports of this function of pseudogenes in section 3.2:

“Through this mechanism, pseudogenes have been showed to influence gene expression in mouse oocytes [45,46] and Trypanosoma brucei [47]. Interestingly, in Tam et al. [45], multiple siRNAs were identified to resemble the sequence from HDAC1; however, most of them were derived from pseudogenes of HDAC1.”

Paragraph 4.3, lines 285-288, sentence unclear, please make it clearer.

ANS: The line was meant to indicate that one pseudogene form the study, namely a pseudogene form PPM1K, was able to regulate multiple coding genes. The line has been edited as follows:

“Among these pseudogenes, a particular one originated from protein phosphatase 1K, mitochondrial (ψPPM1K), was identified as having the potential to regulate multiple protein-coding genes.”